# Fe(III) Biomineralization in the Surface Microlayer of Acid Mine Waters Catalyzed by Neustonic Fe(II)-Oxidizing Microorganisms

Javier Sánchez-España [1,*], Andrey M. Ilin [2], Iñaki Yusta [2], Charlotte M. van der Graaf [3] and Irene Sánchez-Andrea [4]

1 Department of Geological Resources, IGME-CSIC, Calera 1, 28760 Tres Cantos, Spain
2 Department of Geology, Faculty of Science and Technology, University of the Basque Country (UPV/EHU), Apdo. 644, 48940 Bilbao, Spain; andrey.ilin@ehu.eus (A.M.I.); i.yusta@ehu.eus (I.Y.)
3 Department of Biology, University of Cádiz, Puerto Real, 11510 Cádiz, Spain; lot.vandergraaf@uca.es
4 Laboratory of Microbiology, Wageningen University & Research, Stippeneng, 46708 Wageningen, The Netherlands; irene.sanchezandrea@wur.nl
* Correspondence: j.sanchez@igme.es

**Abstract:** The formation of thin mineral films or encrustations floating on the water surface of low-flow or stagnant zones of acid mine drainage (AMD)-affected streams is probably among the most exotic features that can be found in mining areas. However, most fundamental questions about their origin (biotic vs. abiotic), structure, mineralogy, physical stability and metal-retention capacity remain unanswered. This study aims to reveal the factors promoting their formation and to clarify their composition in detail. With this purpose, the major mineral phases were studied with XRD in surface film samples found in different mine sites of the Iberian Pyrite Belt mining district (SW Spain), and the major oxide and trace metal concentrations were measured with XRF and/or ICP-MS. Fe(III) minerals dominated these formations, with mineralogy controlled by the pH (jarosite at pH~2.0, schwertmannite at pH 2.5–3.5, ferrihydrite at pH > 6.0). Other minerals have also been identified in minor proportions, such as brushite or khademite. These mineral formations show an astounding capacity to concentrate, by orders of magnitude ($\times 10^2$ to $\times 10^5$), many different trace metals present in the underlying aqueous solutions, either as anionic complexes (e.g., U, Th, As, Cr, V, Sb, P) or as divalent metal cations (e.g., Cu, Zn, Cd, Pb). These floating mineral films are usually formed in Fe(II)-rich acidic waters, so their formation necessarily implies the oxidation of Fe(II) to Fe(III) phases. The potential involvement of Fe(II)-oxidizing microorganisms was investigated through 16S rRNA gene amplicon sequencing of water underneath the Fe(III)-rich floating mineral films. The sequenced reads were dominated by *Ferrovum* ($51.7 \pm 0.3\%$), *Acidithiobacillus* ($18.5 \pm 0.9\%$) and *Leptospirillum* ($3.3 \pm 0.1\%$), three well-known Fe(II)-oxidizing genera. These microorganisms are major contributors to the formation of the ferric mineral films, although other genera most likely also play a role in aspects such as Fe(III) sequestration, nucleation or mineral growth. The floating mineral films found in stagnant acidic mine waters represent hotspots of biosphere/hydrosphere/atmosphere interactions of great value for the study of iron biogeochemistry in redox boundaries.

**Keywords:** acidophilic microorganisms; microbial iron oxidation; neuston; acid mine drainage; biomineralization; metal/microbe/mineral interaction

## 1. Introduction

Neustonic microorganisms are those inhabiting the water/atmosphere boundary, the so-called surface microlayer (SML) in water bodies such as streams, lakes, ponds or the ocean [1–5]. Due to its elevated surface tension the surface microlayer accumulates organic compounds, fatty acids, proteins and polysaccharides derived from the slow decomposition of organic matter from dust particles, pollen, fallen tree leaves, etc. The dense layer of

organic compounds often leads to the development of a slick with an oily appearance, providing a stable and nutritionally advantageous microenvironment due to a higher nutrient availability (e.g., organic compounds, $O_2$, $H_2$, $CO_2$) [6]. Favorable conditions usually lead to cell densities (cells/volume) in the surface microlayer several orders of magnitude higher than that in the bacterioplankton living a few cms below [1–5,7]. However, the SML represents an extreme environment with high salinity (due to strong evaporation during the summer), high water temperature and elevated exposure to harmful ultraviolet (UV) radiation. Thus, the neustonic microbial community inhabiting the surface microlayer can be significantly different than the planktonic communities living deeper in the water column and may include microorganisms specially adapted to these conditions [1,2,5,8]. According to certain studies, adaptation of the neustonic microorganisms may even include a special resistance to certain antibiotics [9].

However, despite its high scientific interest, the bacterioneuston has been investigated to a much lesser extent compared to the bacterioplankton, and the knowledge of its microbial ecology is mainly limited to oceanic research [1–4]. Therefore, while much research has been undertaken on the microorganisms inhabiting the water column and sediments of acidic streams and lakes affected by acid mine drainage (AMD) [10–22], there is only a single study on the microbial populations forming large filaments on the water surface of an acidic pond in a mining area of the Tinto river basin [8]. Cloning and sequencing of the 16S rRNA gene indicated that these macrofilamentous growths included (mostly heterotrophic) bacteria that did not closely affiliate with species commonly found in the water of acidic streams and ponds of the area (e.g., *Acidithobacillus*, *Leptospirillum*, *Acidiphilium* [10]). Rather, sequences corresponded to microorganisms closely related to *Pseudomonas*, *Caulobacter*, *Sphingomonas*, *Ralstonia*, *Bacillus* and *Halomonas*. This suggests that the surface microlayer of acidic, metal-rich streams, ponds and lakes can host highly specialized microbial communities particularly adapted to those singular physico-chemical conditions. Furthermore, we know very little about the chemistry, mineralogy and trace metal content of the floating mineral films (FMF) ("slicks") that are commonly found in low-flow or stagnant acidic mine waters such as those reported in the Tintillo river and other acidic, metal-rich streams of the Iberian Pyrite Belt (IPB) mining area in SW Spain (Figure 1) [23,24]. These mineral coatings are likely formed by the interaction of neustonic microorganisms of the surface microlayer with aqueous and gas species existing at both sides of this transitional layer, though these particular films have not been the subject of specific research. In short, their chemical and mineralogical nature are still poorly defined, and the geomicrobial interactions leading to their formation are still unknown.

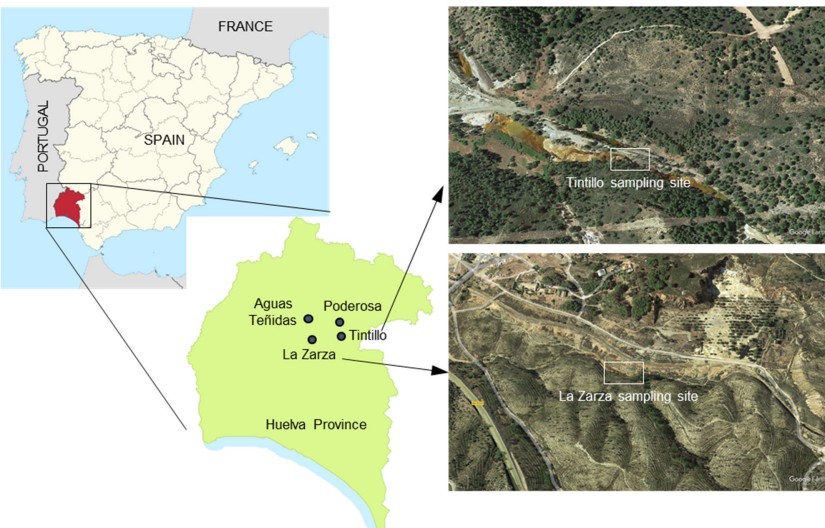

**Figure 1.** Location of sampling points for this study and satellite (Google Earth) images of Tintillo and La Zarza sampling sites.

In this paper, the major geochemical and mineralogical characteristics of the FMFs are reported. The physical, chemical and geomicrobiological factors that seem to control the formation of FMFs in AMD-affected environments are also discussed. We base our study on observations obtained during several years in different acid mine drainage settings of the IPB. Our results show a fundamental role of Fe(II)-oxidizing bacteria and a large metal enrichment in this singular type of iron biomineralization.

## 2. Materials and Methods

### 2.1. Field Work and Sampling

The floating mineral films were sampled during different field campaigns conducted between March 2003 and July 2020. These floating mineral films were always found near the source of acidic mine waters emerging from different mine sites in the province of Huelva (Figures 1 and 2). The acidic leachates included effluents from waste-rock piles (e.g., headwaters of the Tintillo acidic stream [37°32′44″ N, 6°37′41″ W]), mine tunnels (e.g., La Zarza–Perrunal mine [37°42′21″ N, 6°51′56″ W], Poderosa mine [37°44′55″ N, 6°39′22″ W]) and underground mine environments (e.g., Aguas Teñidas mine [37°46′56″ N, 6°50′11″ W]). Samples of floating mineral films were directly collected with clean spatulas (Figure 2B) and introduced in zip-lock plastic bags and/or Eppendorf vials. These samples were stored on ice in a cooling box and transported to the laboratory for further analyses.

The physico-chemical parameters (pH, temperature, redox potential, electric conductivity) were recorded with specific electrodes (e.g., pH meter and redox meters from Hanna) or multiparametric probes (e.g., YSI, Hach [Loveland, CO, USA]) in the aqueous solution in close contact (1–2 cm below) with the sampled FMFs. All electrodes were previously calibrated with fresh buffers and standards supplied from Hanna (Eibar, Spain). Aliquots of water samples (125 mL) were also collected for chemical analyses of major ions and trace elements with 60 mL syringes. The samples were filtered on site with a Merck-Millipore manual filtration unit and 0.45 μm nitrocellulose membrane filters (Merck Millipore, Burlington, MA, USA). Filtered water was stored in 125 mL polyethylene bottles, acidified with HCl (1 M) and kept at 4 °C until chemical analysis in the laboratory. Additional samples from the Tintillo acidic stream (bottle with 500 mL volume) and from several acidic pit lakes (Filón Centro and Guadiana, both in buckets of 4.4 L volume) were also collected and stored in polyethylene bottles for Fe(II) oxidation and FMF formation experiments in the lab. The Tintillo samples were preserved at different pH conditions (natural sample at pH 3.0 and $H_2SO_4$-acidified sample at pH 2.2) to evaluate the effect of pH on the FMF mineralogy.

Samples for microbial community analysis were taken from the Tintillo acidic stream (Figures 1 and 2) [23] in order to compare the results with previous microbiological studies in La Zarza–Perrunal effluent [25]. The biomass was obtained by filtration of the water (~300–400 mL) collected in the layer below sites of FMF formation. The samples were filtered over ~0.3 μm glass fiber filters (Advantec Grade G75 glass fiber filters, 47 mm; Dublin, CA, USA) pretreated as indicated in [21]. Filters were stored at −20 °C until laboratory analysis.

### 2.2. Chemical and Mineralogical Analyses

Once in the lab, the FMF samples were washed with Milli-Q water to prevent secondary salt formation, then air-dried at ambient temperature and stored in closed Eppendorf vials until analyses. A small part of each sample (~5–10 g) was left for detailed mineralogical analyses using X-ray diffraction (XRD) and/or scanning electron microscopy (SEM). The rest of the sample was ground in an agate mortar mill for chemical analyses. Major elements (Fe, Al, Ca, Mg, Mn, Na, K, Si, S) were analyzed with X-ray fluorescence (XRF) using a PANalytical MagiX instrument (Malvern Panalytical Ltd, Malvern, UK). Trace elements were measured with either inductively coupled plasma-atomic emission spectrometry (ICP-AES; Agilent 7500ce, Agilent Technologies, Las Rozas, Madrid, Spain)

(Cu, Zn) and/or inductively coupled plasma-mass spectrometry (ICP-MS; Varian Vista MPX, Varian, Palo Alto, CA, US) (As, Cd, Co, Cr, Ni, Pb, Se, Th, U).

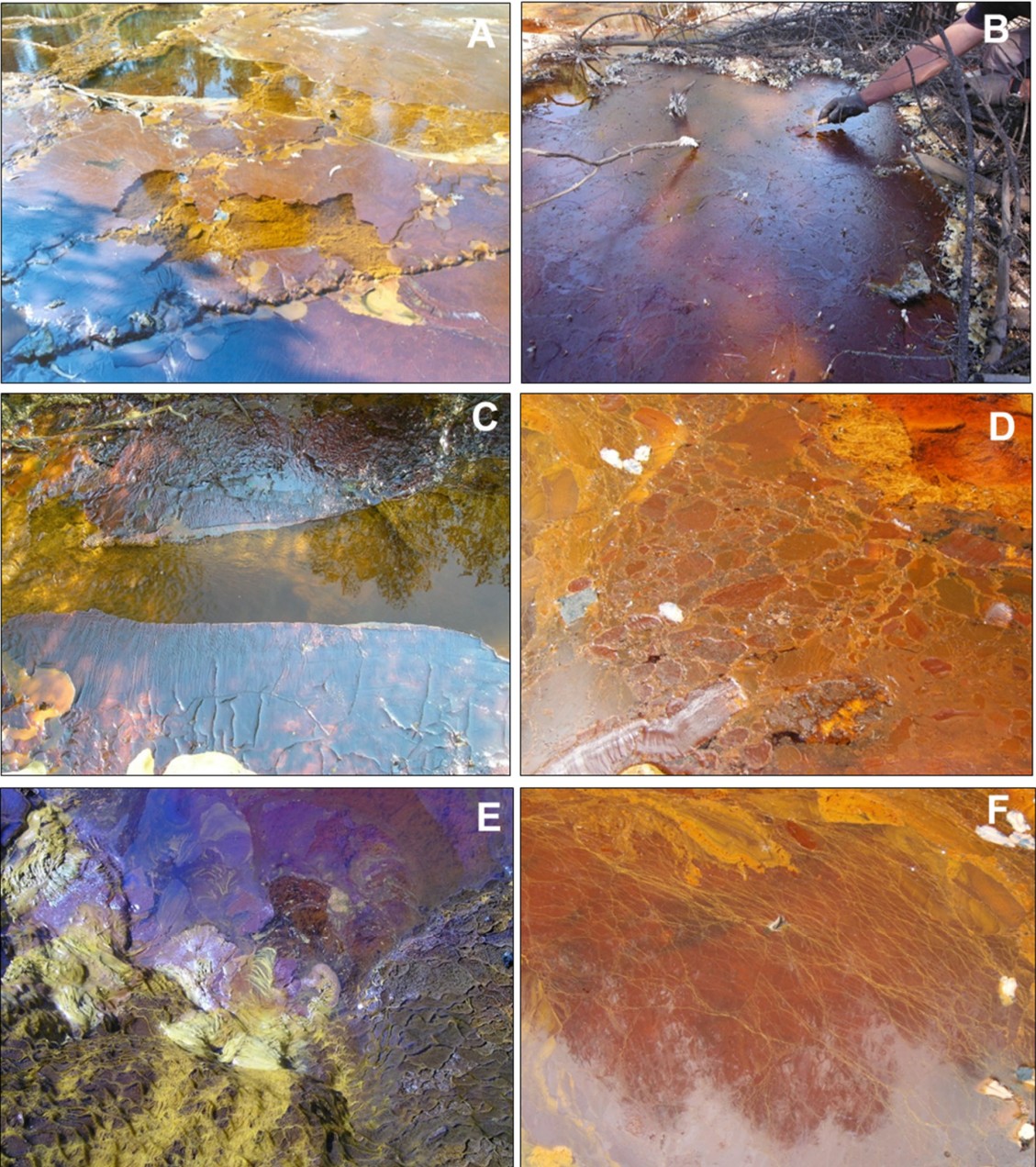

**Figure 2.** Pictures showing diverse textural features of floating mineral films (FMFs) found at the water/atmosphere interface in stagnant or low-flow acidic mine waters of the IPB mining district (Huelva, SW Spain): (**A**) FMF formed in the acidic Tintillo stream near Corta Atalaya in Riotinto mine; note the partial fragmentation by the action of wind; (**B**) Thick FMF formed in a marginal zone near (**A**) that was sampled in May 2019; the surface tension of this FMF covering the whole area allowed accumulation of efflorescent sulfate salts and fallen pine needles; (**C**–**F**) FMFs formed in the acidic effluent emerging from a mine tunnel in La Zarza–Perrunal mine (Huelva, SW, Spain): (**C**) small folds indicate wind-driven shear stress and the unmineralized central zone marks the presence of very low water flow; (**D**) Breccia-like texture formed by accretion of different generations of films growing in various directions; (**E**) Bacterial streamers anchored to pebbles and in close contact with an FMF; (**F**) FMF showing abundant, long, yellow filaments connected and physically stabilized by the iron mineralization.

The concentrations of element species in the water samples were measured with atomic absorption spectrometry (AAS) (Na, K, Ca, and Mg) using a Varian SpectrAA 220 FS (Palo Alto, CA, USA), inductively coupled plasma-atomic emission spectrometry (ICP-AES) (Al, Cu, Fe, Mn, $SO_4$, $SiO_2$, and Zn) using an Agilent 7500ce (Agilent Technologies, Las Rozas, Madrid, Spain), or inductively coupled plasma-mass spectrometry (ICP-MS) (As, Cd, Co, Cr, Ni, Pb, Se, Th, U) using a Varian Vista MPX instrument (Palo Alto, CA, US). The detection limit was 1 mg/L for major elements and 1 μg/L for trace elements.

FMF samples from the Tintillo acidic stream were examined with a JEOL JSM-7000F field emission scanning electron microscope (SEM) coupled with an Oxford INCA 350 energy-dispersive X-ray spectrometry (EDS) detector at the SGIker Facilities (UPV/EHU) (JEOL, Tokyo, Japan; Oxford Instruments, Abingdon, UK). The working conditions included an acceleration voltage of 20 kV, a beam current of 1 nA and a 10 mm working distance. Samples were transferred as a suspension with Milli-Q water onto carbon tape adhered to the carbon holder. After 4 min of plasma cleaning, it was coated with a 15 nm thick carbon layer in a Quorum Q150T ES turbo-pumped sputter coater (Quorum Technologies, East Sussex, UK).

*2.3. Microbial Community Analysis*

DNA extractions and PCR amplification were performed according to van der Graaf et al. [21]. Paired-end reads were processed using NG-Tax version 2.0 [26] on the Galaxy platform at https://www.systemsbiology.nl/ngtax/ (accessed on 17 April 2020), as described previously [21]. ESVs were classified according to the SILVA SSU rRNA database (v128) [27]. ESVs were exported in the .biom format and analyzed further with R (v 4.1.0) [28] in RStudio (2022.12.0 + 353) [29] using the Microbiome [30] and Phyloseq [31] packages. The sequences have been deposited in European Nucleotide Archive (ENA) at EMBL-EBI under accession number PRJEB59480.

## 3. Results

*3.1. Configuration and Temporal Evolution of the FMFs*

The four sampled FMFs were found near the discharge points of acidic leachates emerging from either waste-rock piles (Figure 2A,B) or mine tunnels (Figures 2C–F and 3). These emerging acidic waters are usually characterized by the common AMD characteristics (i.e., low pH, high sulfate and metal concentrations [10–25]) and, in addition, by a near-absence of dissolved oxygen (see next section) and very high concentrations of dissolved ferrous iron ($Fe^{2+}$) [23]. The FMFs were only observed in pools or marginal zones of the streams with very low flow or stagnant water, suggesting that these mineral films need close-to-motionless conditions to allow mineral growth on the water surface. In certain cases, the mineral films appeared cracked or slightly rumpled, possibly by the action of wind or slight water motion (Figure 2A,C). The appearance of the four FMFs found at the mine sites showed common features, including: (1) a significant thickness (~1 mm) of the FMF above the water surface, (2) a significant superficial extension that can cover several square meters of the water surface in a given part of the streams, (3) a dark red color, which is usually related to the presence of Fe(III) minerals, and (4) an elevated surface tension, which allows the FMFs to remain on the water surface and accumulate salts, fallen tree leaves or plant debris without disruption of the FMF (Figure 2B). In some cases, the FMFs may also show certain green tones, presumably due to the presence of neustonic green microalgae [23] and it is sometimes possible to see gas bubbles trapped in the FMF (*not shown*). On other occasions, the FMFs may exhibit a breccia-like internal arrangement by the accretion of different, small-scale encrustations growing in different directions (Figure 2D). This brecciated textural pattern has also been observed in FMFs formed in the lab at the water/atmosphere boundary in flasks filled with originally anoxic acidic water from Tintillo stream (Figure 3A,B) and with waters from the surface from two acidic pit lakes (Figure 3C,D).

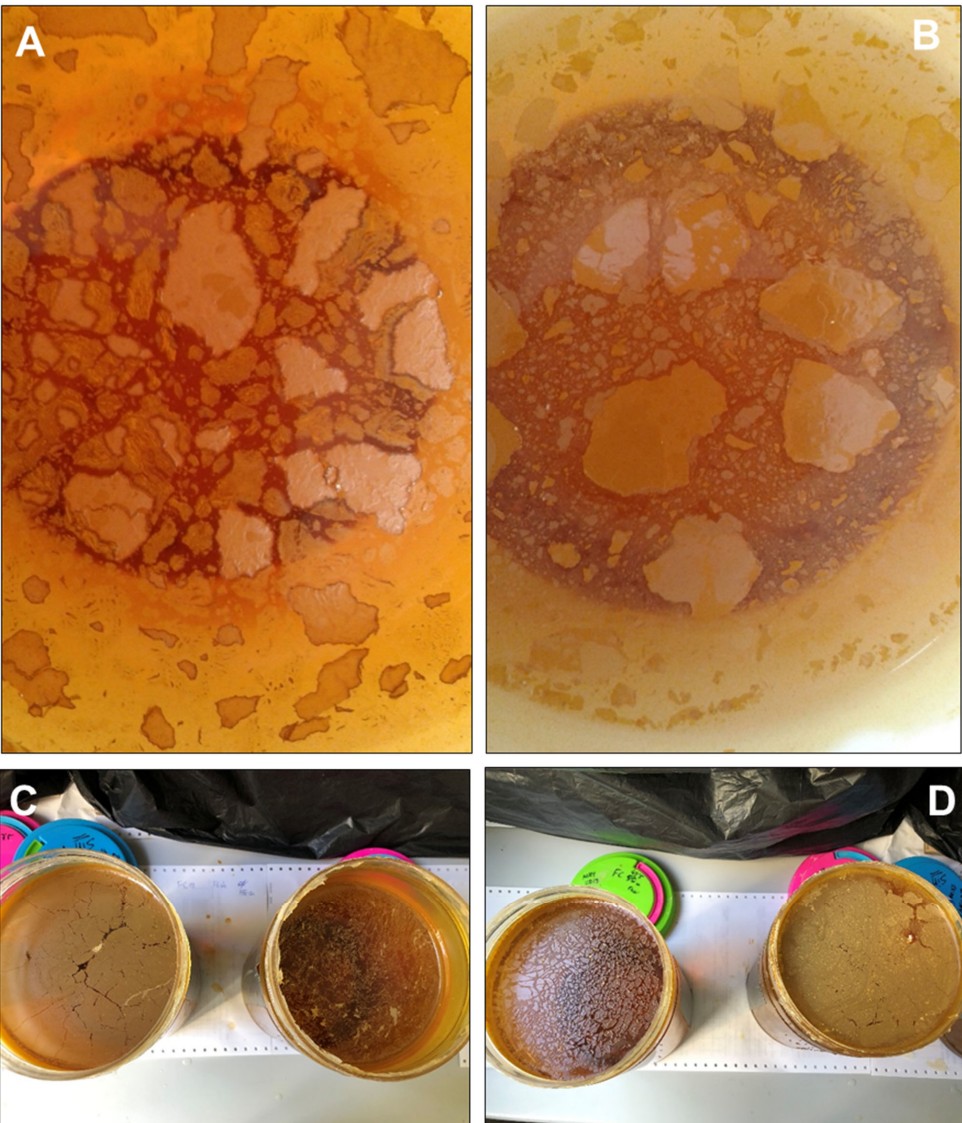

**Figure 3.** Floating mineral films formed on stagnant acidic mine water samples from the acidic Tintillo stream at pH 3.0 (**A**) and pH 2.2 (**B**) and from acidic pit lakes (Guadiana, (**C**); Filón Centro, (**D**)) at pH 2.1. These samples were stored in glass flasks (14 cm in diameter) to assess the formation of floating mineral films under laboratory conditions; note the ocellated or breccia-like textures observed in many of the flasks, in which the individual fragments exhibit lighter cores and darker rims (suggestive of different oxidation extent and/or degree of mineralization). The field of view in A–B is 10 cm across.

In these laboratory samples, it was also possible to observe "fossilized" bubbles due to the entrapment and subsequent iron mineralization of a gas phase (likely consisting of $CO_2$) in the FMF (see Section 3.3). The presence of streamers in close contact with the FMFs (Figure 2E) or long filaments dispersed across the mineral coating (Figure 2F) reveals microbial activity in or around these FMFs.

The thickness and surface extension of the FMFs, however, are not constant in time, as they can evolve seasonally and may experience important changes even on a diurnal time frame (Figure 4). Certain acidic streams such as the effluent emerging from the La Zarza–Perrunal mine portal have been observed to be completely devoid of any FMF in the summer season (Figure 4A) and entirely covered by a thick and bulging FMF in the winter (Figure 4B). These seasonal fluctuations suggest that physical parameters such as the temperature (which has a strong influence on oxygen diffusion in water and in the

water/atmosphere gas exchange rate [32,33]) are likely important parameters indirectly controlling FMF formation. Furthermore, in different settings (e.g., the same portal effluent in La Zarza–Perrunal mine) the FMF present in the morning has been observed to partially disappear in the afternoon (Figure 4C,D), suggesting partial redissolution of the films. In the absence of detailed studies, it could be speculated that these diurnal changes result from the photoreduction of Fe(III) minerals in the FMF, which is a well-known process and has already been reported in the study area [34].

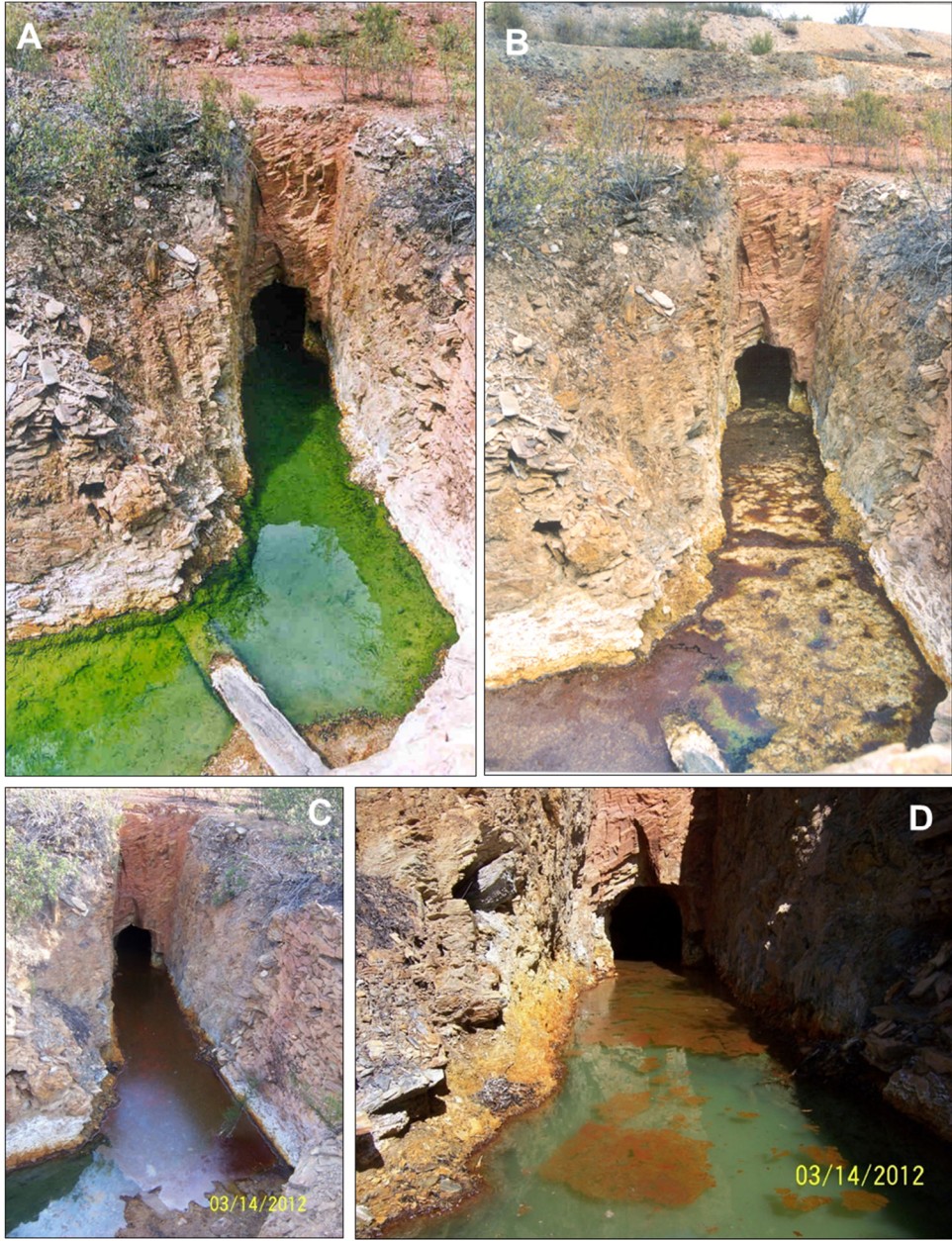

**Figure 4.** Pictures showing temporal changes of the floating mineral films (FMFs) formed at the water/atmosphere boundary in the acidic effluent emerging from the Perrunal mine portal (Huelva, SW, Spain): (**A**) summer 2003, (**B**) winter 2003, (**C**) 14 March 2012 at 10:30 am, and (**D**) 14 March 2012 at 1:30 pm. Note the clear difference in the surface development and thickness of the FMFs at different time scales: seasonal (winter vs. summer situation; (**A**,**B**)) and diurnal (morning vs. afternoon situation; (**C**,**D**)). Picture A is reproduced from [35] with kind permission from Springer. Pictures in (**C**,**D**) are courtesy of Dr. William D. Burgos (Pennsylvania State University).

The formation of FMFs has also been observed in underground mine settings, where these mineral formations can develop singular ocellated or "leopard-skin-like" textures (Figure 5). There, the FMFs have not only been observed covering the water surface in pools of partly flooded mine tunnels (Figure 4A) but also covering the surface of wet walls of the mine galleries (Figure 5C,D) and even on sulfide-rich muds in the gallery floor (Figure 5B). The ocellar texture is apparently formed by orange ocelli surrounded by dark brownish red rims (Figure 5), which suggests a kind of mineralogical evolution (see later section).

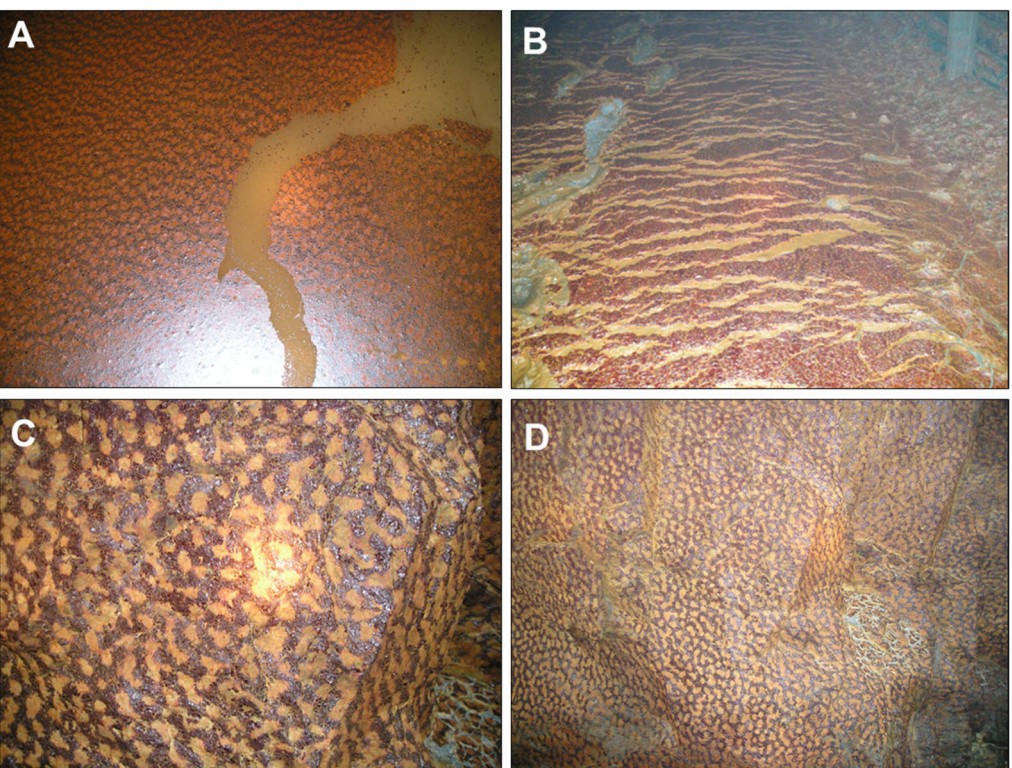

**Figure 5.** Iron-rich mineralized films found in the underground Aguas Teñidas mine (Huelva, SW Spain): (**A**) floating mineral film found in stagnant water in a mine tunnel; (**B**) mineral coating formed on water-saturated, pyrite-rich sediments deposited on the ground of a mine tunnel; (**C,D**) mineral coatings formed on wet gallery walls. The mineralogy of these biofilms was always dominated by ferrihydrite with minor amounts of other phases such as natrojarosite and brushite (see Section 3.3). Note the ocellated or "leopard-skin"-like texture of the biofilms in (**A,C,D**).

### 3.2. Field Parameters and Aqueous Composition of Underlying Waters

The field parameters measured on site and the chemical composition of waters underlying the sampled FMFs are given in Table 1. A common feature of all waters developing FMFs on the water surface was that they were in all cases highly oxygen-deficient to nearly anoxic, with oxygen concentrations ranging from 0.1 to 1.6 mg/L $O_2$. These data are low even if compared to the average AMD effluent in the IPB (Table 1). Since the common concentrations in $O_2$-saturated mine waters of the area are usually between 6.5 and 9.5 mg/L $O_2$ at 20–25 °C [35,36], these data roughly represent around 2 to 20% oxygen saturation. Other common features of these FMF-forming waters were: (1) relatively low redox values (Eh = 533 − 595 mV) typical of reduced, Fe(II)-rich waters [36] and (2) Fe(II)-dominated waters, representing up to 71–89% of the total iron content (Table 1).

Most waters, except the Aguas Teñidas mine groundwater, were also acidic (pH 1.6–3.1) and sulfate- and metal-rich (9060–24,760 mg/L $SO_4^{2-}$, 357–2046 mg/L Al, 20–302 mg/L Cu, 56–557 mg/L Zn, 53–17,300 µg/L As; Table 1). This is in line with the chemical compositions usually observed in mine waters of the study area (Table 1) and result from intense sulfide oxidation and mineral dissolution [37]. The case of Aguas

Teñidas, an underground sulfide mine, is somehow exotic since this water was near-neutral (pH 6.6) and showed a very low redox value (70 mV) and very low sulfate and metal concentrations in comparison with the rest of the FMF-forming waters (Table 1). These unusual geochemical features of the Aguas Teñidas mine water may be derived from an interaction with carbonate minerals present in the wall rocks of the mine galleries, although this possibility cannot be demonstrated and is beyond the scope of this study. However, although at trace concentrations, this water did also contain some residual dissolved ferrous iron (0.92 mg/L) indicating very low $O_2$, in addition to aluminum and zinc (5 mg/L Al, 4.1 mg/L Zn).

**Table 1.** Selected physico-chemical parameters and metal concentrations measured in waters co-existing with the studied floating mineral films (all values refer to single measurements in the sampling points).

| Sample | T | DO | pH | Eh | Fe(II) | Fe(III) | $SO_4^{2-}$ | Al | Cu | Zn | As |
|---|---|---|---|---|---|---|---|---|---|---|---|
| | °C | mg/L | | mV | mg/L | mg/L | mg/L | mg/L | mg/L | mg/L | µg/L |
| Tintillo | 24.5 | 1.6 | 2.7 | 541 | 1205 | 495 | 24,700 | 1810 | 184 | 557 | 525 |
| Perrunal | 25.6 | 1.0 | 3.1 | 533 | 2323 | 300 | 9060 | 357 | 20 | 56 | 2254 |
| Poderosa | 22.1 | 0.1 | 1.6 | 595 | 2100 | 560 | 10,400 | 431 | 302 | 149 | 17,300 |
| AT | 25.0 | 0.5 | 6.6 | 70 | 0.92 | b.d. | 370 | 5 | 0.05 | 4.1 | 2 |
| Exp. film | 24.2 | 0.7 | 2.7 | 572 | 1230 | 450 | 24,760 | 2046 | 132 | 447 | 53 |
| AMD IPB [1] | 19.0 | 4.6 | 2.7 | 624 | 801 | 476 | 7440 | 386 | 64 | 169 | 2123 |

[1] Average value for 53 different AMD effluents measured in the IPB (taken from [37]). Abbreviations: AT, Aguas Teñidas; DO, dissolved oxygen; Eh, redox potential referred to the standard hydrogen electrode; Exp. film, experimental film; b.d., below detection.

### 3.3. Mineralogical and Chemical Composition of the Floating Mineral Films

The mineralogical analyses of the floating mineral films with XRD showed different mineral phases (Figure 6). The mineral encrustations were always dominated by Fe(III) minerals such as schwertmannite [$Fe^{3+}_{16}O_{16}(OH)_{12}(SO_4)_2$; Figure 6B] and/or jarosite [$(K,Na)Fe^{3+}_3(SO_4)_2(OH)_6$; Figure 6E], although some other minerals were also identified at minor proportions, including gypsum ($CaSO_4 \cdot 2H_2O$; Figure 5B), khademite [$Al(SO_4)F \cdot 5(H_2O)$; Figure 6E] and brushite [$Ca(HPO_4) \cdot 2(H_2O)$]; Figure 6G) ]. Khademite and brushite are rarely described in AMD environments [24,37,38], being typical of either strongly evaporative conditions (khademite) or cave systems rich in organic substances (brushite).

**Table 2.** Major oxide concentrations (measured with XRF and given in wt.%) of iron-rich floating mineral films found in IPB mine waters. All values refer to single measurements conducted on representative samples from each site.

| Sample | $SiO_2$ | $Al_2O_3$ | $Fe_2O_3$ | CaO | $K_2O$ | MgO | $Na_2O$ | $P_2O_5$ | $SO_3$ | LOI |
|---|---|---|---|---|---|---|---|---|---|---|
| Tintillo | 1.78 | 1.35 | 58.40 | 0.11 | 0.10 | 0.41 | 0.24 | 0.20 | n.a. | 37.40 |
| Perrunal | 0.33 | 0.40 | 60.01 | 0.40 | 0.30 | 0.32 | <0.10 | 0.20 | 15.70 | 38.40 |
| Poderosa | 10.65 | 6.66 | 10.81 | 2.00 | 5.00 | 1.90 | 1.35 | 14.60 | 46.28 | n.a. |
| AT | 4.99 | 1.49 | 59.10 | 1.10 | 0.14 | 0.58 | 0.22 | 0.05 | n.a. | 32.40 |
| Exp. film | 0.70 | 1.83 | 52.05 | 0.92 | 0.10 | 0.57 | 0.03 | 0.80 | 15.62 | 41.74 |

Abbreviations: AT, Aguas Teñidas; LOI, Lost on Ignition; n.a., not analyzed.

It is worth noting that the XRD technique is mostly suitable for detecting fairly crystalline phases, so that the poorly crystalline to amorphous phases that may coexist in the FMF samples are difficult to detect. For example, schwertmannite and ferrihydrite ($Fe^{3+}_2O_3 \cdot 0.5(H_2O)$), the two most common and typical Fe(III)-minerals found in AMD settings [35–41], were not automatically detected with the XRD software but manually recognized by identification of their most intense broad peaks in the diffractograms (Figure 6B,D). The presence of the wide humps around 4.80 Å, 3.40 Å, 2.53 Å, 1.93 Å, 1.66 Å and 1.51 Å is indicative of schwertmannite [40], while ferrihydrite humps might be localized

at 2.59 Å, 2.25 Å, 1.98 Å, 1.73 Å and 1.46 Å [42]. The comparison of the diffractogram from Aguas Teñidas mine (Figure 6G) with the available reference data suggests the presence of 6-line ferrihydrite. This is in line with the high pH and low sulfate content of the parent solution, while schwertmannite is usually the dominant phase in acidic and high-sulfate environments [35–40]. In addition, the chemical composition of the FMFs obtained using XRF (Table 2) showed iron concentrations around 58–60 wt.%, matching those commonly found in schwertmannite or ferrihydrite-rich samples [35–41]. This suggests that these minerals are actually the most common in some of the samples (e.g., schwertmannite in Tintillo and Perrunal, ferrihydrite in Aguas Teñidas).

**Table 3.** Trace element concentrations (given in ppm and measured with ICP-MS and/or ICP-AES) of iron-rich floating mineral films found in stagnant mine waters of the IPB (Figures 2–5). All values refer to single measurements conducted on representative samples from each site.

| Sample | As | Cd | Cr | Cu | Pb | Zn | U | Th |
|---|---|---|---|---|---|---|---|---|
| Tintillo | 709 | 3 | 14 | 421 | 297 | 383 | 650 | 325 |
| Perrunal | 6166 | 1267 | 231 | 55 | 26 | 211 | 180 | 77 |
| AT | 8179 | 64 | 1 | 5381 | 200 | 14,172 | n.a. | n.a. |
| Exp. film | 88 | 24 | 155 | 348 | 12 | 507 | 1 | 3 |
| Sed. IPB [1] | 943 | n.a. | 17 | 1150 | 321 | 911 | n.a. | n.a. |
| MDM IPB [2] | 1664 | 5 | 18 | 813 | 222 | 751 | n.a. | n.a. |

[1] Average value for natural sediments in AMD-affected streams of the Iberian Pyrite Belt (taken from [37]);
[2] Average value for iron-rich mine drainage minerals (MDM) found in mine sites of the Iberian Pyrite Belt (taken from [37]). Abbreviations: AT, Aguas Teñidas; n.a., not analyzed.

This was confirmed with SEM investigations of FMFs formed in laboratory experiments with originally anoxic water from the Tintillo acidic stream and two acidic pit lakes (Figure 3). The SEM pictures obtained from these FMFs showed that schwertmannite (with a characteristic *pin-cushion* or *hedgehog* texture) and jarosite (as pseudohexagonal crystals) were both very common (Figure 7). Schwertmannite was the dominant mineral in the FMF formed on waters at pH 3.0, whereas jarosite was abundant in the FMF formed on waters at pH 2.2. The comparison of the XRD results with the physico-chemical parameters reported in the previous section (Table 1) suggests that the mineralogy of the FMFs is strongly controlled by the pH, with jarosite found at pH~2.0 (Poderosa), schwertmannite at pH 2.5–3.5 (Tintillo, Perrunal) and ferrihydrite at pH > 6.0 (Aguas Teñidas). This finding matches observations made in AMD environments worldwide [24,35–41]. The presence of schwertmannite has been reported in FMFs observed in different AMD settings [24,43,44]. This mineral is closely associated with FMF formation at the air/water interface when Fe(II)-rich, $O_2$-poor AMD is in contact with the oxygenic atmosphere and water flow is slow enough to allow mineral precipitation at the SML.

The trace element concentrations of the FMFs obtained with ICP-MS or ICP-AES (Table 3) also indicate that FMFs may have an astounding capacity to concentrate many different trace metals present in the underlying aqueous solutions either as divalent metal cations (e.g., $Cu^{2+}$, $Zn^{2+}$, $Cd^{2+}$, $Pb^{2+}$) or as anionic complexes (e.g., $UO_2(SO_4)_2^{2-}$, $ThO_2(SO_4)_2^{2-}$, $H_2AsO_4^-$, $HCrO_4^-$, $H_2VO_4^-$, $SbO_4^{3-}$, $PO_4^{3-}$). The concentrations of some metals (e.g., As, Cd and Cr in Perrunal, or As, Cu and Zn in Aguas Teñidas) are notably high even if compared to the average values for natural sediments and Fe-rich, mine drainage minerals of AMD-affected streams of the IPB (Table 3). The calculated enrichment factors using the metal concentrations measured in the FMFs (Table 3) and in the corresponding parent solutions (Table 1) are on the order of $\times 10^2$ to $\times 10^5$, indicating an extreme metal enrichment in the mineral films (Table 4).

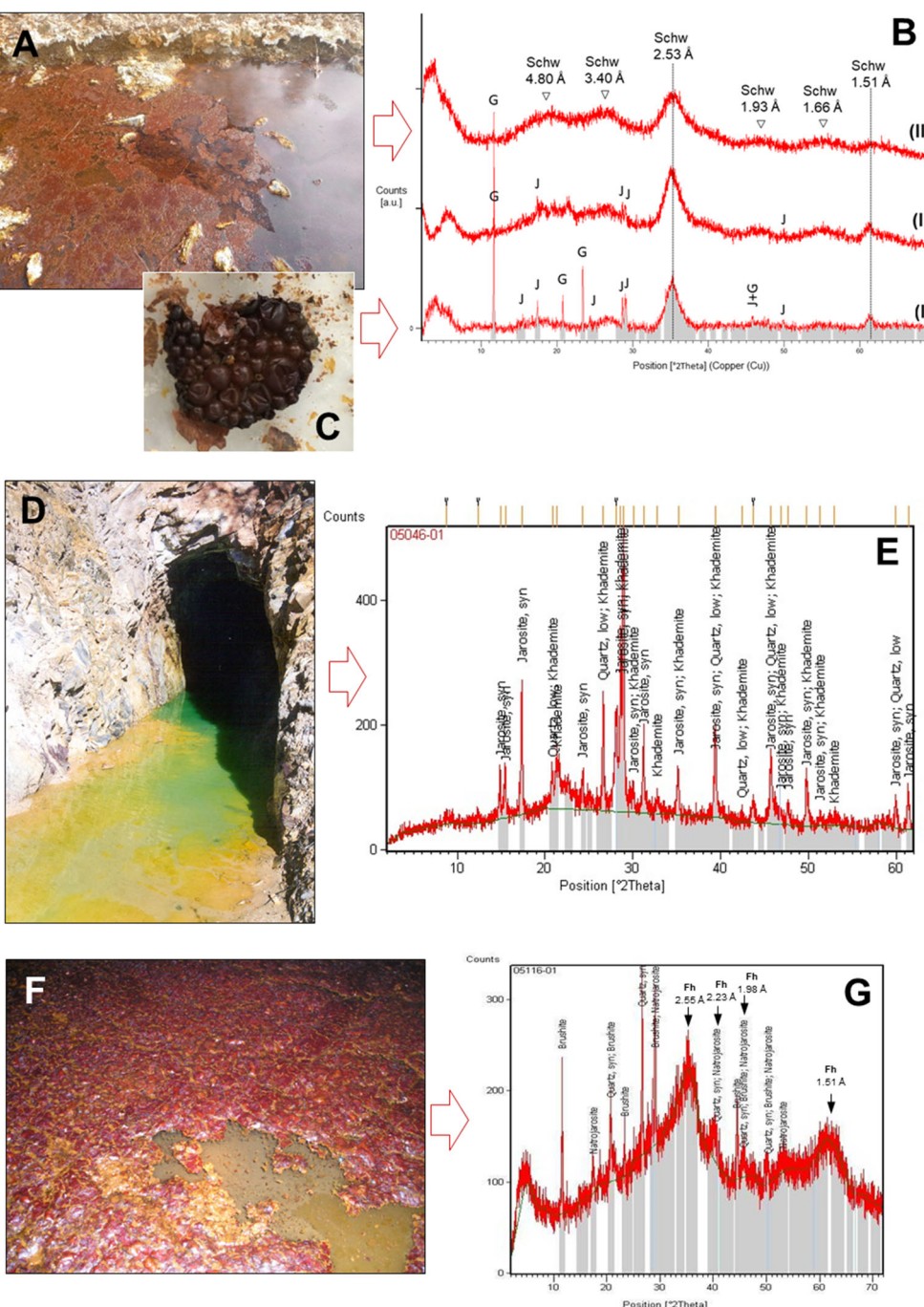

**Figure 6.** Examples of major mineral phases identified with XRD in samples of floating mineral films (FMFs) found on the surface of different mine drainage sites of the IPB (Huelva, SW Spain): FMFs formed in Tintillo acidic stream (**A**), Poderosa mine portal (**D**) and an underground gallery of Aguas Teñidas mine (**F**), along with their corresponding XRD patterns (**B**,**E**,**G**) showing the characteristic humps or peaks of the major mineral phases identified (schwertmannite (schw), jarosite (J) and gypsum (G) in the Tintillo SMF, B; jarosite and khademite in Poderosa, E; natrojarosite, brushite and ferrihydrite –Fh– in Aguas Teñidas, G). See also Tables 2 and 3 for the major oxide and/or trace element composition of these FMFs. The detail in (**C**) shows molds of fossilized bubbles produced by iron (schwertmannite) precipitation around gases (likely, bacterially produced $CO_2$) trapped at the water/atmosphere boundary in a flask containing Tintillo acidic water and stored for months in the lab.

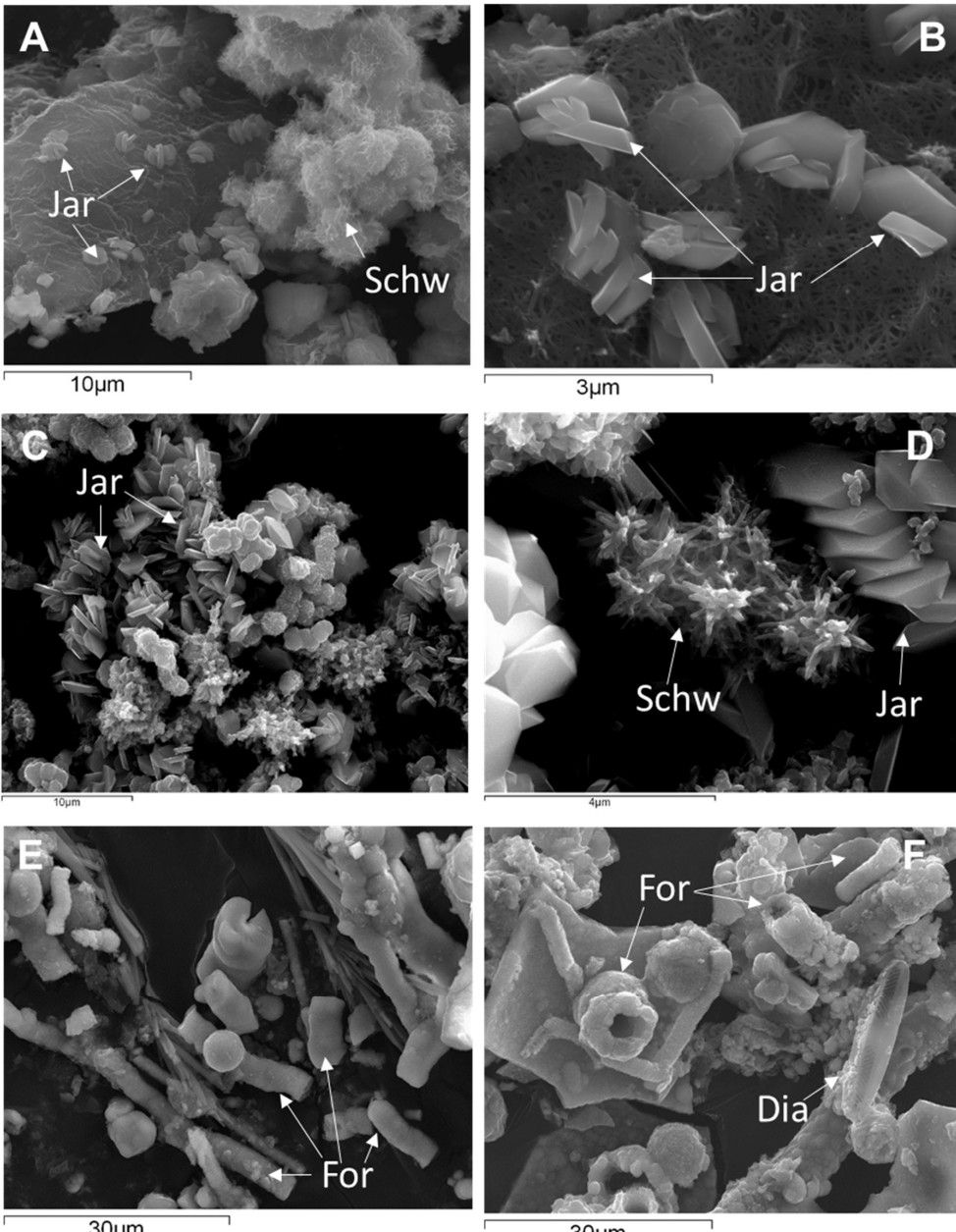

**Figure 7.** SEM pictures obtained from floating mineral films formed in lab experiments with originally anoxic waters from the Tintillo acidic stream (**A**,**B**,**E**,**F**) and Guadiana acidic pit lake (**C**,**D**). (**A**,**D**) show schwertmannite (Schw) with characteristic *pin-cushion* or *hedgehog* texture and formed at pH~3.0 coexisting with euhedral crystals of jarosite (Jar) formed at pH~2.2. (**E**,**F**) show abundant fossilized organic remnants (For) and sparse diatoms (Dia) in the sample from the Tintillo stream.

Another interesting feature found in the SEM study is the presence of abundant fossilized organic remnants (FORs) and sparse photosynthetic algae and diatoms (Figure 6E,F) which denote the presence of microscopic life in these FMFs. Many organic remnants consisting of elongated particles with a central conduct are considered to have formed by iron mineralization of algal filaments or fungal hyphae (Figure 7E,F).

### 3.4. Microbiology of FMF-Forming Waters

The microbiological analyses of the waters underlying the floating mineral film of the Tintillo acidic stream performed through 16S rRNA gene amplicon sequencing showed that the two genera *Ferrovum* (51.7 ± 0.3%) and *Acidithiobacillus* (18.5 ± 0.9%) dominated the

sequenced reads (Figure 8; Table 4). Species in both genera are well-known Fe(II)-oxidizers typical of AMD environments [10–19,22] and previously found to be the dominant Fe(II)-oxidizers in the La Zarza–Perrunal mine portal effluent [25], as well as in the nearby Tinto river [10]. In addition, other genera known for Fe(II) oxidation, all typical for AMD waters, were detected in lower relative abundance, including *Leptospirillum*, *Acidibacillus*, *Acidithrix* and *Thiomonas*. In the genera *Metallibacterium*, *Acidibacter* and *Acidiphilium*, also detected in the sequenced reads, so far only Fe(III)-reducing species have been reported [45–47].

**Table 4.** Compilation of Fe(II)-oxidizing microorganisms, iron-rich minerals and enriched metals identified in FMF-forming acidic mine waters of the IPB. Microorganisms and minerals in bold are dominant, whereas those in italics are accessory.

| Mine Site (Creek) | Description | Fe(II)-Oxidizing Genera and/or Species Identified | Minerals Identified | Enriched Trace Elements (EF) | Ref. Microb. |
|---|---|---|---|---|---|
| Riotinto mine (Tintillo creek) | Seepage from waste-rock pile | ***Ferrovum*** ***Acidithiobacillus*** *Leptospirillum* | **Schw** | U ($\times 650$) Th ($\times 600$) | This study |
| La Zarza–Perrunal (Perrunal creek) | Effluent from mine portal | ***At. Ferrooxidans*** ***Leptospirillum spp.*** ***Ferrovum sp.*** *Xanthomonas* *Ferritriphicum sp.* *Ferrimicrobium sp.* *Thermoplasmata* | **Schw** | As ($\times 434$) Cr ($\times 10^3$) V ($\times 443$) U ($\times 10^3$) Th ($\times 10^4$) Cd ($\times 10^3$) Cu ($\times 10^3$) Zn ($\times 10^3$) | 25 |
| Poderosa mine | Effluent from mine portal | n.a. | **Jar**, *Kd* | P ($\times 10^5$) | |
| Aguas Teñidas | Stagnant water in mine tunnel | n.a. | **Fh**, *Br, Na-Jar* | As ($\times 10^6$) Cu ($\times 10^2$) Zn ($\times 3.5$) Pb ($\times 10^5$) P ($\times 10^3$) | |
| Experimental biofilm | Tintillo water stored in glass flask | Water from Tintillo creek analyzed above | **Schw, Jar** | Zn ($\times 1$) V ($\times 170$) Cr ($\times 30$) As ($\times 200$) Sb ($\times 170$) | This study |

Abbreviations: EF, enrichment factor (with respect to concentration in water); n.a., not analyzed. Ref. Microb., reference for microbiological data of Fe(II)-oxidizing microorganisms: Minerals: Schw, schwertmannite; Jar, jarosite; Na-Jar, natrojarosite; Fh, ferrihydrite; Br, brushite; Kd, khademite.

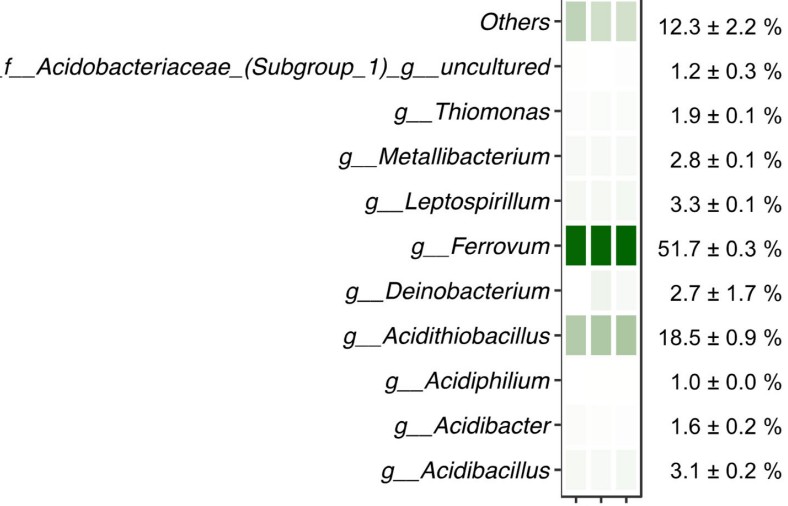

**Figure 8.** Microbiological analysis of water underlying a floating mineral film in Rio Tintillo showing relative abundance of sequenced reads assigned to taxa indicated on the left; f__: family, g__: genus. Relative abundance is shown per replicate on a color scale (white–green representing minimum to maximum relative abundance, respectively). Average relative abundance is indicated on the right.

## 4. Discussion

### 4.1. Role of Fe(II)-Oxidizing Microorganisms in the Formation of Floating Mineral Films

The reported findings confirmed the initial hypothesis of the FMFs being mainly composed of Fe(III) minerals such as schwertmannite or jarosite (Figure 5, Table 4). The first and most obvious implication of the presence of these Fe(III) oxyhydroxysulfates on the surface of Fe(II)-rich waters is that the formation of the FMFs likely requires the participation and metabolic activity of Fe(II)-oxidizing microorganisms to catalyze the oxidation of Fe(II) to Fe(III) for mineral precipitation. Therefore, the FMFs observed in these mine sites likely represent a particular case of iron biomineralization by neustonic acidophiles that concentrate in the surface microlayer to find more favourable conditions for their Fe(II)-oxidizing metabolisms. The potential geomicrobiological interactions leading to FMF formation are represented in Figure 9. The element of particular interest for these microorganisms would be molecular oxygen, which is very scarce in the bulk aqueous phase (microaerobic conditions, usually <1.6 mg/L $O_2$) but much more abundant in the SML due to diffusion from the atmosphere. In addition, bacteria migrating to the SML could also benefit from a higher availability of nutrients such as phosphate, nitrate or $CO_2$, which could be more abundant near the water surface due to the decomposition and partial dissolution of organic matter (e.g., fallen tree leaves, atmospheric deposition). This possibility, however, needs to be demonstrated and requires the use of microelectrodes capable of establishing vertical gradients of $O_2$ and/or nutrient concentration at mm scale.

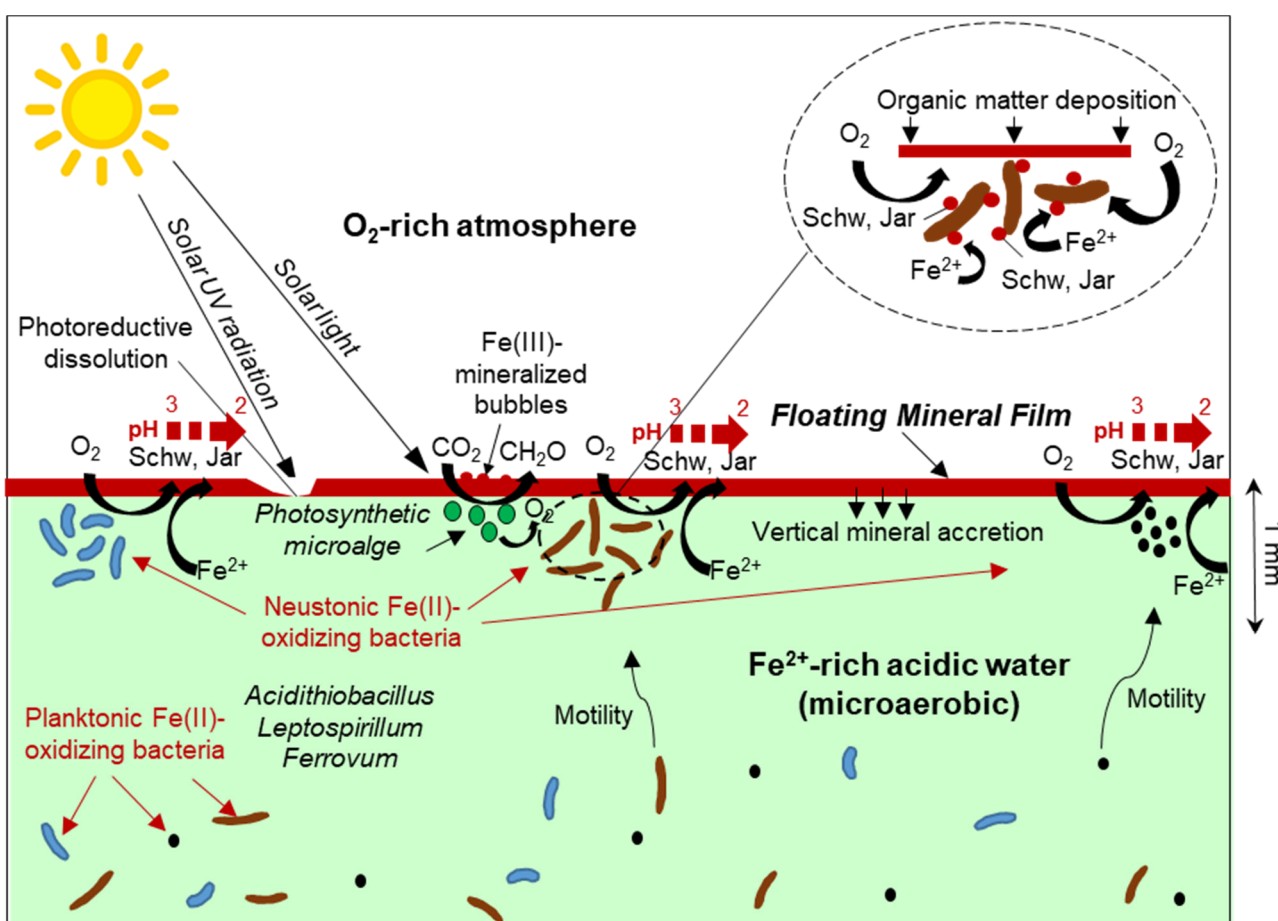

**Figure 9.** Schematic representation of metal/microbe/mineral interactions leading to Fe(III)-rich floating mineral film formation in acid mine drainage settings. The inset shows a magnification of Fe(II)-oxidizing bacteria inhabiting the surface microlayer and catalyzing the precipitation of Fe(III) minerals at or near the cell wall. Schw: Schwertmannite, Jar: jarosite.

The concept of planktonic bacteria moving to the surface microlayer to find a high availability of nutrients and organic carbon has been observed in the ocean and in freshwater systems (lakes, rivers), where they form dense neustonic communities adapted to the specific physico-chemical conditions of the SML [1–7,48–50]. The presence of surface slicks with an oily sheen in fresh waters (see for example [51,52]), resembling the sheen observed in Figure 1E, has been attributed to the presence of Fe(II)-oxidizing bacteria such as *Leptothrix discophora* and the resulting precipitation of iron hydroxide [52,53], though these slicks had been the subject of little chemical or mineralogical research [54]. In AMD and other extreme environments, neustonic communities have rarely been subject to specific research, and we are only aware of one study in the Tinto River that observed marked differences in the microbial composition of the neustonic community of a pool with respect to that found in the water column [10]. However, in other systems no significant differences between the neustonic and the planktonic communities were found, suggesting that the neustonic communities may be formed by microorganisms moving from the underlying water by chemotaxis and motility [1,2,48–50].

The composition of the neustonic community in the studied FMFs cannot be established with certainty because a specific microbiological sampling of these micron-sized layers was not conducted. However, the high relative abundance of common Fe(II)-oxidizing genera in the 16S rRNA gene amplicon sequencing reads from the nearly anoxic waters sampled in the Tintillo (this study) and the previously studied microbial community at La Zarza–Perrunal [25] streams where the FMF form allow us to speculate about the colonization of the SML by these bacteria. This process would imply obvious energetic and metabolic advantages of the much higher oxygen availability near the water/atmosphere interface. This higher oxygen availability for the oxidation of Fe(II) would not only be derived from diffusion from the atmosphere, but could also be favored by the presence of neustonic green microalgae, which can produce elevated $O_2$ concentrations by photosynthesis [2] (Figure 9). The presence of green patches in some of the FMFs suggests that this photosynthetic $O_2$ production could be indirectly contributing to increase the Fe(II) oxidation kinetics in the SML. The widespread presence of spherical structures likely represents Fe(III)-mineralized bubbles of photosynthetically produced oxygen trapped in the SML and mineralized by Fe(III) oxyhydroxysulfates (e.g., schwertmannite) at the water/air boundary [44,55]. However, these structures could also have been formed by the mineralization of bacterially produced $CO_2$, as observed in our laboratory experiments (Figure 5C). In any case, the oxidation rate of Fe(II) in the SML would be much higher than in the underlying water not only because of the higher oxygen availability (which is a rate-determining factor in these environments), but also because of a higher concentration of biomass and cell density of Fe(II)-oxidizing bacteria in this layer [1–7].

The precipitation of minerals such as schwertmannite, jarosite or ferrihydrite (depending on the pH conditions and aging time) on the water surface would be favoured by the elevated surface tension of the pre-existing neustonic biofilms developed in the SML, which could also provide nucleation sites in the form of bacterial cells, algal filaments or microbial exudates (polysaccharides, proteins, etc.) (Figure 7E,F and Figure 9) [8,10]. From this perspective, these microbial–mineral aggregates may represent exotic geomicrobial formations of particular geobiological interest due to the metal/microbe/mineral interactions involved in their formation.

The internal configuration and texture of the studied FMFs is also a remarkable feature that seems to be related to the strong dependence of Fe(II) oxidation and Fe(III) precipitation on bacterial activity. The textural features observed in the FMFs in the field and in the lab (e.g., ocelli or leopard-skin-like textures, filaments intergrown with the Fe(III) minerals; Figures 4–6) suggest that this Fe(III) precipitation is likely not a purely physico-chemical process (e.g., like the bulk precipitates formed by abiotic Fe(II) oxidation in AMD waters [35–37]) but microbially controlled. However, this hypothesis requires further demonstration in future studies. The precipitation of Fe(III) and the formation of FMFs, at the same time, would represent an advantage for the neustonic microbial

community, which could benefit from a more rigid and solid substrate in which organic carbon and nutrients could be more abundant than in the aqueous phase. In addition, the FMF may act as a "mineral shield" protecting the neustonic bacteria from harmful UV radiation, as described for mineralized cyanobacteria in hot springs [56].

In any case, it could be speculated that the formation of FMFs in the acidic streams depends on a delicate equilibrium of many different factors, which may include: (1) bacterial oxidation kinetics, (2) the rate of downwards oxygen diffusion into the water column, (3) the rate of upwards diffusion of ferrous iron to the SML, (4) the water flow rate, (5) the evaporation rate, (6) the solar radiation intensity, and (7) the wind speed, to name a few.

### 4.2. Unusual Mineralogy and Metal Enrichment in the FMFs Implying Singular Physico-Chemical Conditions

Another interesting feature found in the studied FMFs is that these mineral accretions may include exotic minerals rarely found in AMD settings and typical of either highly evaporative environments (e.g., khademite, an aluminum sulfate) or of cave systems rich in phosphate (e.g., brushite, a calcium phosphate) (Figure 5, Table 4). The presence of brushite in the FMF of the Aguas Teñidas mine implies the abundance of phosphate in this mine water (which is coherent with observations in most mine waters of the area [37]), whereas the presence of khademite (reported in some places as a post-mining efflorescence [24,57]) may be considered diagnostic of strongly evaporative conditions in standing waters containing high concentrations of dissolved aluminum and sulfate. Thus, the khademite–jarosite mineral assemblage characterizing the FMF formed in the effluent of the Poderosa mine (Figure 5) is diagnostic of strongly acidic and evaporative conditions. On the other hand, the abundance of ferrihydrite in the thick FMF found in Aguas Teñidas is coherent with the circumneutral pH of this water (Table 1) [54]. The detection of traces of natrojarosite in this sample was somehow surprising and may indicate that the conditions of FMF formation may have evolved or experienced certain oscillations from acidic to neutral (or vice versa) as a natural response to either microbial activity (e.g., enhanced sulfide oxidation in adjacent sediments) or mixing with more acidic water in the mine tunnel.

A final comment about the ferric minerals forming the FMFs refers to their remarkable capacity to concentrate, by orders of magnitude, different trace metal(oid)s present in the aqueous phase. Some of these elements have been frequently reported as being adsorbed on poorly crystalline schwertmannite in AMD settings of the IPB [35–40]. However, some others, such as U or Th, which have been shown to be highly enriched in wetlands in association with organic matter [58], are very rarely found in mine drainage ferric precipitates [59]. This may suggest that the singular precipitation conditions of ferric iron in the FMFs could be enhancing the adsorption of these ionic species onto the mineral surfaces of poorly crystalline schwertmannite. However, the mechanisms favouring this uranium and thorium entrapment in the microbially formed FMFs (which could include many different processes such as biosorption, bioaccumulation or chelation with microbial exudates or microbially driven redox reactions) are still unclear and deserve further study.

### 5. Conclusions

The present study highlights the importance of geomicrobial (metal/microbe/mineral) interactions for the formation of Fe-rich, floating mineral films in acid mine drainage settings. Abiotic factors do play a role in the development of these thin mineral layers on the water surface (e.g., slow water motion, high evaporation and elevated surface tension all tend to enhance mineral precipitation, whereas, on the other hand, strong wind or high solar radiation are destabilizing factors). However, the metabolic activity of neustonic Fe(II)-oxidizing bacteria living in the surface microlayer appears to be a critical factor since they provide the Fe(III) required for mineral precipitation. It is hypothesized that Fe(II)-oxidizing bacteria such as *Ferrovum*, *Aciditopbacillus* or *Leptospiriullm* could obtain clear benefits in colonizing the surface microlayer because of the higher availability of oxygen,

organic matter and nutrients (e.g., nitrate, phosphate, $CO_2$) of this microenvironment with respect to the underlying water. These advantages would compensate the disadvantages of living in an extreme microenvironment characterized by higher ionic concentrations and more intense exposure to harmful solar radiation. The composition and microbial ecology of the neustonic microbial community forming the iron-rich floating mineral films and their differences with respect to the planktonic communities are still to be revealed. However, despite the precise features of the microorganisms catalyzing the oxidation of Fe(II) and the precipitation of Fe(III), it seems clear that the studied FMFs represent a singular case of iron biomineralization that could provide important clues about the biogeochemical cycling of iron in extreme environments.

**Author Contributions:** Conceptualization, J.S.-E.; Methodology, J.S.-E.; Validation, J.S.-E., I.Y., A.M.I., C.M.v.d.G. and I.S.-A.; Formal Analysis, J.S.-E., I.Y., A.M.I., C.M.v.d.G. and I.S.-A.; Investigation, J.S.-E., I.Y., A.M.I., C.M.v.d.G. and I.S.-A.; Resources, J.S.-E., I.Y., I.S.-A.; Data Curation, J.S.-E., A.M.I. and C.M.v.d.G.; Writing—Original Draft Preparation, J.S.-E.; Writing—Review and Editing, J.S.-E., I.Y., A.M.I., C.M.v.d.G. and I.S.-A.; Project Administration, J.S.-E.; Funding Acquisition, J.S.-E., I.Y. and I.S.-A. All authors have read and agreed to the published version of the manuscript.

**Funding:** This research was funded by the Spanish Ministry of Science and Innovation through grant number CGL2016-74984-R to J.S.-E., the research program TTW by the Dutch Research Council (NWO) under project number 14797 to I.S.-A and the Basque Government (Consolidated Group IT1678-22) to I.Y. and A.M.I.

**Data Availability Statement:** The 16S rRNA gene amplicon sequences have been deposited in the European Nucleotide Archive (ENA) at EMBL-EBI under accession number PRJEB59480.

**Acknowledgments:** William Burgos (Pennsylvania State University, State College, PA, USA) and E. López-Pamo (IGME-CSIC, Spain) are thanked for providing some of the field pictures included in this work. We are also grateful to the technical staff at IGME-CSIC for their support during the lab work and chemical analyses of solid and water samples and personnel at the SGIker facilities at UPV/EHU during the XRD and SEM analyses.

**Conflicts of Interest:** The authors declare no conflict of interest. The funders had no role in the design of the study; in the collection, analyses, or interpretation of data; in the writing of the manuscript; or in the decision to publish the results.

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
