# Peer review of "Fe(III) Biomineralization in the Surface Microlayer of Acid Mine Waters Catalyzed by Neustonic Fe(II)-Oxidizing Microorganisms"

_minerals, doi:10.3390/min13040508_

Round 1
Reviewer 1 Report
Review
Fe(III) biomineralization in the surface microlayer of acid mine 2 waters catalyzed by neustonic Fe(II)-oxidizing microorganisms
The submitted article has a certain scientific interest, is well written and illustrated and can be published after minor editing.
Comments
1. Please, avoid to use abbreviations where it is not necessary. Why should the reader strain and memorize all these abbreviations? Try to make it easier for him to understand the material.
Write “surface microlayer“ instead of abbreviation “SML”.
Try to use “floating mineral films” instead of “FMF”, acid mine drainage instead of AMD”.
Lines 80-82. “We also discuss the physical, chemical and geomicrobiological factors that seem to control the formation of FMFs in AMD-affected environments. We base our study on observations obtained during several years in different AMD settings of the IPB”. What is IPB?
2. The references on articles about the neuston, mainly to works published in 1970 - 1999.
Check out the 2021 article Stabnikova, O.; Stabnikov, V.; Marinin, A.; Klavins, M.; Klavins, L.; Vaseashta, A. (2021) Microbial Life on the Surface of Microplastics in Natural Waters. Applied Sciences (Switzerland), 11, 11692. https://doi.org/10.3390/app112411692
Author Response
1. Please, avoid to use abbreviations where it is not necessary. Why should the reader strain and memorize all these abbreviations? Try to make it easier for him to understand the material. Write “surface microlayer“ instead of abbreviation “SML”. Try to use “floating mineral films” instead of “FMF”, acid mine drainage instead of AMD”.
Reply: We just want to avoid repetition of the long terms and keep the manuscript short. The term "floating mineral film" is repeated 92 times along the manuscript, so we consider it is a good idea to avoid using the full term all the time. However, we have revised the text and, whenever possible, we have replaced the abbreviations by the corresponding full names of the different elements discussed in the text.
Lines 80-82. “We also discuss the physical, chemical and geomicrobiological factors that seem to control the formation of FMFs in AMD-affected environments. We base our study on observations obtained during several years in different AMD settings of the IPB”. What is IPB?
Reply: IPB refers to the "Iberian Pyrite Belt". We had just defined this term a few lines above (line 73)
2. The references on articles about the neuston, mainly to works published in 1970 - 1999. Check out the 2021 article Stabnikova, O.; Stabnikov, V.; Marinin, A.; Klavins, M.; Klavins, L.; Vaseashta, A. (2021) Microbial Life on the Surface of Microplastics in Natural Waters. Applied Sciences (Switzerland), 11, 11692. https://doi.org/10.3390/app112411692
Reply. Thanks for the recommendation of this paper, which includes a nice revision of bacterioneuston research. We have added this reference to the References section (we have replaced another reference by Fehon and Oliver which was not so relevant for this study).
Reviewer 2 Report
The paper investigates the geomicrobiological factors that control the formation of floating mineral films (FMFs) in AMD-affected environments. The experimental approach is well designed to address the objectives of the study and the methods adequately described. The findings are well presented, but the interpretation can be improved. My comments are appended below for consideration.
Fig 2 Items around the samples could be distracting, the authors must consider to take the photo in a less congested setting...
The legibility of Fig 5B should be improved
The values in Table 3 Should be expressed in SI unit not ppm.
Page 13, line 394: “…found, suggest-394 ing that the neustonic communities form by microorganisms from the underlying water 395 column via chemotaxis and motility”. The authors must check the sentence.
Fig 8: The consideration of anoxic conditions may not be exact, under anoxic conditions Leptospirillum and Ferrovum will not survive while At will rather reduce Fe(III); microaerobic conditions may be likely in this case where Ferrovum is considered among dominant species. It is important to clearly establish the mechanism of formation of the FMF, which is preceded by the agglomeration of organic matters at the surface of water, before adherence of bacteria to the existing structure and then oxidation Fe(II) to survive, then adsorption of Fe(III) to the existing structure.
The authors must refrain from using the personal pronoun “we”.
Author Response
The paper investigates the geomicrobiological factors that control the formation of floating mineral films (FMFs) in AMD-affected environments. The experimental approach is well designed to address the objectives of the study and the methods adequately described. The findings are well presented, but the interpretation can be improved. My comments are appended below for consideration.
Reply: Thanks very much for your opinion and suggestions.
Fig 2 Items around the samples could be distracting, the authors must consider to take the photo in a less congested setting...
Reply: Unfortunately, these flasks were discarded once the lab experiments had been finished, so we cannot take any better picture.
The legibility of Fig 5B should be improved
Reply: We agree. We have modified this panel. increasing the font size so that the mineral names can be now read in this panel.
The values in Table 3 Should be expressed in SI unit not ppm.
Reply: Trace elements measured in solids are usually given in ppm (or ppb) in most studies of environmental science and mineralogy, so we prefer to maintain this table with concentrations in ppm.
Page 13, line 394: “…found, suggest-394 ing that the neustonic communities form by microorganisms from the underlying water 395 column via chemotaxis and motility”. The authors must check the sentence.
Reply: We have slightly rephrased this sentence to improve its readibility.
Fig 8: The consideration of anoxic conditions may not be exact, under anoxic conditions Leptospirillum and Ferrovum will not survive while At will rather reduce Fe(III); microaerobic conditions may be likely in this case where Ferrovum is considered among dominant species. It is important to clearly establish the mechanism of formation of the FMF, which is preceded by the agglomeration of organic matters at the surface of water, before adherence of bacteria to the existing structure and then oxidation Fe(II) to survive, then adsorption of Fe(III) to the existing structure.
Reply. We agree. We have replaced the term "anoxic" for "microaeroboic", which is probably more correct. Also, we have introduced slight changes in this figure to better illustrate the FMF formation processes.
The authors must refrain from using the personal pronoun “we”.
Reply: We have changed to the passive voice through the entire manuscript, so that now we avouid the use of "we".
Reviewer 3 Report
The Authors present a very interesting work work on authuigenic biologically-mediated formation of minerals in surface layers in water flows form mine areas. The opinion of this reviewer is that the works is highly interesting and original and it certainly deserve to be published in Minerals, or even in a higher impact journal. The work is very well written and structured, and the reading is easy. The manuscript was carefully prepared and the Authors took the necessary time to present their results in a comprehensive way. Methodology is also well described and robust. This reviewer wants to express gratitude to the authors for the well-done work. The recommendation of this reviewer is going to be ACEPT AFTER MINOR REVISON, but only to let the Authors to consider some of my comments and recommendations that will be around formal aspects mainly.
Firstly, I would recommend the Authors to include a final section of “CONCLUSSIONS”. Conclusions are well presented and verbalized in the discussion section, however, a final section of perhaps one paragraph could be useful to find the major findings of the work.
I would also recommend merging figures 1 and 2.
Some minor comments:
.- In the caption of Fig. 1 (L. 85-98) the authors include results (e.g. L.90), this is strange because the figure is far away from the results section. The caption also contains the first mention to a table (Table 3; L. 90), which distorts the numeration of tables. I would recommend not including results not mentions to table 3 in the caption.
.- This reviewer would recommend including a (new) Fig. 1 with a location map. This is interesting as they submitted the manuscript to an international journal. Relevant data in the map could be (i) the study areas at least in the context of the European continent, (ii) the extension of the Iberian Pyrite Belt in the Iberian Peninsula, and (sampling points). I would also recommend including the coordinates of the sampling points within the text.
.- In the methodology (L. 133-140) the Author mention several techniques used to determine the elemental composition of the FMF samples, i.e. XRF, ICP-AES, ICP-MS… Can the Authors mention which elements were determined by each technique?
.- In L. 156-158, when the Authors say, “as described previously (#REF]” one understand that it was previously described in the manuscript while the intention is to refer to a reference. I Wold recommend something like: “DNA extractions and PCR amplification was performed according to van der Graaf et al. [26]. Paired-end reads (…) as described by [21].
.- In L. 246-247, the Authors present oxygen contents in percentages; I understand this is oxygen saturation. I recommend to clarify this.
.- In L. 257-262, Can the Authors explain better this sentence? From “These unusual geochemical features…” to the end of the paragraph.
.- Table 1, I would recommend to include comparison with reference values (e.g. [37]) in table 1. What are the values included in the table? Are they from a single measurement, are them means, medians, maximums… Please, indicate it in the caption. Check it in the other tables.
.- Table 3, Can the Authors include in the table any comparison with general references or references of SPM unaffected by AMD?
.- Review the first paragraph of the discussion (L. 364-381). Could it be placed in a new concussions section?
.- Table 4, in the column of references there are two hyphens. What it means? Same reference than above (i.e. 25) or a different thing? Please, clarify.
.- This reviewer is not sure of the use the Authors do of the term “fossilized bubbles” (e.g. L. 422), I understand the concept but not sure if the term is correct. Please, check it.
.- L. 445. A point is missing after “future studies”.
I want to finish highlighting a very positive point of the paper. The Authors acknowledge the limitations of their work and present them not as debilities but as opportunities, opening questions to be responded in future works. This is something this reviewer appreciate, and I want to encourage the Author to continue in this research line.
Author Response
The Authors present a very interesting work work on authuigenic biologically-mediated formation of minerals in surface layers in water flows form mine areas. The opinion of this reviewer is that the works is highly interesting and original and it certainly deserve to be published in Minerals, or even in a higher impact journal. The work is very well written and structured, and the reading is easy. The manuscript was carefully prepared and the Authors took the necessary time to present their results in a comprehensive way. Methodology is also well described and robust. This reviewer wants to express gratitude to the authors for the well-done work. The recommendation of this reviewer is going to be ACEPT AFTER MINOR REVISON, but only to let the Authors to consider some of my comments and recommendations that will be around formal aspects mainly.
Reply: We are deeply thankful for these kind words, and we sincerely thank the suggestions and recommendations of this reviewer, which will help to increase the final quality of our manuscript
Firstly, I would recommend the Authors to include a final section of “CONCLUSSIONS”. Conclusions are well presented and verbalized in the discussion section, however, a final section of perhaps one paragraph could be useful to find the major findings of the work.
Reply: Thanks. We have added a new section with one paragraph of Conclusions with the main take-home messages of our study.
I would also recommend merging figures 1 and 2.
Reply: We prefer to main them separate, since Fig. 1 (now Fig. 2) refers to field pictures, and Fig. 2 (now Fig. 3) refers to mineral films found in the lab.
Some minor comments:
.- In the caption of Fig. 1 (L. 85-98) the authors include results (e.g. L.90), this is strange because the figure is far away from the results section. The caption also contains the first mention to a table (Table 3; L. 90), which distorts the numeration of tables. I would recommend not including results not mentions to table 3 in the caption.
Reply: We agree. We have deleted this information from the caption of Figure 1.
.- This reviewer would recommend including a (new) Fig. 1 with a location map. This is interesting as they submitted the manuscript to an international journal. Relevant data in the map could be (i) the study areas at least in the context of the European continent, (ii) the extension of the Iberian Pyrite Belt in the Iberian Peninsula, and (sampling points). I would also recommend including the coordinates of the sampling points within the text.
Reply: We have included a new figure in the revised version. The new Figure 1 now provides the location of the sampling points and we provide aereal views of the two most important sampling sites (Tintillo and La Zarza). We also give the coordinates of the sampling points in the text. Regarding the IPB, there are hundreds of papers on the Iberian Pyrite Belt mining district, so we think it is not necessary to indicate the extension of this district in the map.
.- In the methodology (L. 133-140) the Author mention several techniques used to determine the elemental composition of the FMF samples, i.e. XRF, ICP-AES, ICP-MS… Can the Authors mention which elements were determined by each technique?
Reply: We now provide this information in the Methods section.
.- In L. 156-158, when the Authors say, “as described previously (#REF]” one understand that it was previously described in the manuscript while the intention is to refer to a reference. I Wold recommend something like: “DNA extractions and PCR amplification was performed according to van der Graaf et al. [26]. Paired-end reads (…) as described by [21].
Reply. We have changed this sentence, as suggested.
.- In L. 246-247, the Authors present oxygen contents in percentages; I understand this is oxygen saturation. I recommend to clarify this.
Reply: Yes. In any case, we have slightly changed this sentence to make it clearer (adding "saturation").
.- In L. 257-262, Can the Authors explain better this sentence? From “These unusual geochemical features…” to the end of the paragraph.
Reply. We refer to the fact that they near-neutral pH of this mine water is probably the result of the interaction with acidic water with acid-neutralizing minerals (such as carbonates) present in the wall rocks. We have slightly changed this sentence to gain clarity.
.- Table 1, I would recommend to include comparison with reference values (e.g. [37]) in table 1. What are the values included in the table? Are they from a single measurement, are them means, medians, maximums… Please, indicate it in the caption. Check it in the other tables.
Reply: OK. We have modified Tables 1 and 3 to include this information for reference (Table 2 just provides major oxides and we think it's not necessary to modify this table). The values refer to single measurements in the sampling points. We now indicate this in the corresponding table captions of the three tables.
.- Table 3, Can the Authors include in the table any comparison with general references or references of SPM unaffected by AMD?
Reply: We now include this information in Table 3, and also in Table 1.
.- Review the first paragraph of the discussion (L. 364-381). Could it be placed in a new concussions section?
Reply: We prefer to maintain this paragraph here to avoid disturb the original line of reasoning of our Discussion section. But we have included a new Conclusions section with a final paragraph highlighting the main messages of our study.
.- Table 4, in the column of references there are two hyphens. What it means? Same reference than above (i.e. 25) or a different thing? Please, clarify.
Reply: We justed wanted to express that these mine sites do not have geomicrobiological information about bacterial species composition (so the hypens mean "no study"). But we have deleted them in the revised version.
.- This reviewer is not sure of the use the Authors do of the term “fossilized bubbles” (e.g. L. 422), I understand the concept but not sure if the term is correct. Please, check it.
Reply: We have changed "fossilized" for "Fe(III)-mineralized", which is probably more correct (or less controversial)
.- L. 445. A point is missing after “future studies”.
Reply: corrected.
I want to finish highlighting a very positive point of the paper. The Authors acknowledge the limitations of their work and present them not as debilities but as opportunities, opening questions to be responded in future works. This is something this reviewer appreciate, and I want to encourage the Author to continue in this research line.
Reply: We really appreciated these kind words, and we thank again this reviewer for his/her valuable suggestions. Of course, our aim is to keep working in this exciting research line and establish with more precision the microorganisms inhabiting the surface microlayer of these AMD sites and their exact role in Fe(III) biomineralization.